# Testing a digital system that ranks the risk of unplanned intensive care unit admission in all ward patients: protocol for a prospective observational cohort study

James Malycha,[1,2,3] Oliver C Redfern,[1,2] Guy Ludbrook,[3] Duncan Young,[1,2] Peter J Watkinson[1,2]

¹Nuffield Department of Clinical Neurosciences, University of Oxford, Oxford, UK
²Kadoorie Centre for Critical Care Research and Education, Oxford University Hospitals NHS Foundation Trust, Oxford, Oxfordshire, UK
³Faculty of Health and Medical Sciences, University of Adelaide, Adelaide, South Australia, Australia

**Correspondence to**
Dr James Malycha;
james.malycha@ndcn.ox.ac.uk

## ABSTRACT

**Introduction** Traditional early warning scores (EWSs) use vital sign derangements to detect clinical deterioration in patients treated on hospital wards. Combining vital signs with demographics and laboratory results improves EWS performance. We have developed the Hospital Alerting Via Electronic Noticeboard (HAVEN) system. HAVEN uses vital signs, as well as demographic, comorbidity and laboratory data from the electronic patient record, to quantify and rank the risk of unplanned admission to an intensive care unit (ICU) within 24 hours for all ward patients. The primary aim of this study is to find additional variables, potentially missed during development, which may improve HAVEN performance. These variables will be sought in the medical record of patients misclassified by the HAVEN risk score during testing.

**Methods** This will be a prospective, observational, cohort study conducted at the John Radcliffe Hospital, part of the Oxford University Hospitals NHS Foundation Trust in the UK. Each day during the study periods, we will document all highly ranked patients (ie, those with the highest risk for unplanned ICU admission) identified by the HAVEN system. After 48 hours, we will review the progress of the identified patients. Patients who were subsequently *admitted* to the ICU will be removed from the study (as they will have been correctly classified by HAVEN). Highly ranked patients *not admitted* to ICU will undergo a structured medical notes review. Additionally, at the end of the study periods, all patients who had an unplanned ICU admission but whom HAVEN *failed to rank highly* will have a structured medical notes review. The review will identify candidate variables, likely associated with unplanned ICU admission, not included in the HAVEN risk score.

**Ethics and dissemination** Approval has been granted for gathering the data used in this study from the South Central Oxford C Research Ethics Committee (16/SC/0264, 13 June 2016) and the Confidentiality Advisory Group (16/CAG/0066).

**Discussion** Our study will use a clinical expert conducting a structured medical notes review to identify variables, associated with unplanned ICU admission, not included in the development of the HAVEN risk score. These variables will then be added to the risk score and evaluated for potential performance gain. To the best of our knowledge,

## Strengths and limitations of this study

► The study methodology is in accordance with the STrengthening the Reporting of OBservational studies in Epidemiology guidelines.
► We describe a method that combines risk score testing with a structured medical notes review conducted by a clinical expert for the iterative improvement of a digital system that quantifies risk for unplanned intensive care unit admission in all ward patients.
► To the best of our knowledge, this is the first study of this type.

this is the first study of this type. We anticipate that documenting the HAVEN development methods will assist other research groups developing similar technology.

**Trial registration number** ISRCTN12518261

## BACKGROUND
### Introduction
Early warning score (EWS) systems, such as the National Early Warning Score, combine abnormalities in patient vital signs into an aggregate score.[1] This score triggers a clinical response when a threshold is exceeded. Despite wide-scale adoption of EWS systems, significant clinical patient deterioration on hospital wards still occurs.[1 2] Additionally, high numbers of false alerts lead to alert 'fatigue' and inefficient use of response teams.[3] Adding additional clinical information to such systems, such as laboratory results and comorbidities, improves specificity.[4–12] However, identifying and adding new variables requires a systematic approach to avoid needless complexity.[13]

We have developed a system to predict the risk of unplanned intensive care unit (ICU) admission (within 24 hours) for patients on general medical and surgical wards. It is

called the Hospital Alerting Via Electronic Noticeboard (HAVEN).[14] To identify potential variables for inclusion in HAVEN, we used a modified Delphi process and a systematic literature review.[15] Those identified variables that were available within the electronic patient record (EPR) were extracted from data sets comprising all patients admitted to two National Health Service (NHS) trusts (a trust is a legal entity that provides goods and services for the purposes of the provision of hospital, community and/or other aspects of patient care).[12] We then used a machine learning method[16] to select the optimal combination of variables for the HAVEN risk score. In contrast to EWS systems, HAVEN was not designed to produce alerts. Instead, HAVEN provides a list of patients in the hospital, ranked from most to least at risk of requiring ICU admission. The intent is that HAVEN will improve patient safety by informing the use of clinical response teams.

### Aims and objectives

The primary aim of this study is to discover additional candidate variables, not recognised during the data-driven derivation process, that would improve the performance of the HAVEN risk score. We will review the medical records of *misclassified* patients, that is, patients ranked highly by HAVEN but who were not admitted to the ICU; or patients who were never ranked highly by HAVEN but had an unplanned ICU admission.

### The HAVEN risk score

The HAVEN risk score is calculated using both *static* and *dynamic* variables extracted in real time from the EPR.

Static variables refer to patient-level data available at admission: age, gender, comorbidities (classified according to the Elixhauser comorbidity index[17]) and Hospital Frailty Risk Score.[18] As diagnostic coding in the UK occurs after a patient has been discharged, the comorbidity index and frailty scores are calculated using a patient's admissions over the previous 2 years. Score performance in patients with no previous admissions (and potentially undocumented comorbidities) will be evaluated separately.

Dynamic variables refer to measurements taken repeatedly during hospital admission, that is, laboratory results and vital signs. The HAVEN risk score is currently updated according to the most recent measurements of: albumin, bilirubin, C reactive protein, haemoglobin, platelets, white cell count, potassium, sodium, urea, creatinine, heart rate, systolic blood pressure, respiratory rate, body temperature, a neurological status assessment using either the Alert-Verbal-Painful-Unresponsive scale or the Glasgow Coma Scale, peripheral oxygen saturation from pulse oximetry ($SpO_2$) and the estimated fraction of inspired oxygen.[19] A patient's HAVEN score is re-calculated each time a new dynamic variable is received by the system and the score is further adjusted for the time since hospital admission.

## METHODS

The study will be reported according to the STrengthening the Reporting of OBservational studies in Epidemiology guidelines.[20]

### Design and setting

This is a prospective, observational, cohort study conducted at the John Radcliffe Hospital, part of the Oxford University Hospitals NHS Foundation Trust in the UK. The John Radcliffe Hospital is a tertiary hospital with over 800 beds and serves a population of over 650 000 people, who are generally more affluent and with higher life expectancy than the national average.[21]

## DATA COLLECTION

Data collection will occur during 4, full, non-consecutive weeks in 2019. The notes review will be undertaken by a senior critical care physician. Patients who are discharged or die during the study period will have these details recorded. They will remain in the analysis data set.

### Participants
#### Eligibility criteria

Emergency and elective adult patients (16 years or over) admitted to medical, surgical, observational or short-stay wards will be eligible for inclusion. We will exclude patients for whom a score cannot be generated (ie, those with no recorded vital sign or laboratory measurements).

#### Sample size

We will sample two subgroups of patients:
1. False High Rank (FHR).
2. False Low Rank (FLR).

The FHR group will consist of patients ranked highly by HAVEN but who were not admitted to the ICU. To identify this group, we will record the five highest-ranked patients on the HAVEN system at 09:00 each morning of the study. After 48 hours, we will remove any patients who were subsequently admitted to the ICU. The remaining patients' records will be reviewed.

The FLR group will be identified at the end of the study and consist of all patients who had an unplanned ICU admission during the study period and were not present in any of the daily high-ranking groups. These patients' records will also undergo a medical notes review.

The study will run for 4 non-consecutive weeks with expected recruitment of between 130 and 150 patients.

### Structured medical notes review

We will carry out a structured review of patient medical notes (electronic and paper-based) for the two sample groups described in section Sample size. From these, we will construct a medical summary, looking specifically at patient-centred and system-based variables associated with decisions around ICU admission. We will use a modified version of the Hogan *et al* qualitative note review techniques.[22] We will then conduct a thematic analysis of the extracted data.[23] It is expected that from within the

themes the additional variables will be identified. Along with the *as yet unknown* variables, the following data will be extracted:

1. Primary diagnosis.
2. Comorbidities and medical history (where not available from previous admissions).
3. Any treatment limitations put in place and the reasons for these, including 'Do not attempt resuscitation' documents.
4. Current medication.
5. Radiological imaging.
6. Point-of-care blood gas analysis.
7. Clinical Frailty Score.[24]

## Qualitative methods

Qualitative data (eg, information in free text) will be analysed thematically, using methods of constant comparison.[25] A coding framework will be constructed to assist in understanding of the data. We will use NVivo software (QSR International, www.qsrinternational.com) to support the qualitative analysis process.

## Patient safety and public involvement

As an observational study of patient records with no intervention, adverse events related to research interventions are not possible. In the event that inadequate care is identified during the structured medical note review, local NHS trust protocols will be followed. Reviewers will act in accordance with the General Medical Councils Good Medical Practice Guidelines (2013). This action includes acting immediately if a patient is not receiving basic care to meet their needs. If patients are at risk because of inadequate premises, equipment or other resources, and policies or systems, we will correct the matter if possible and raise our concerns in line with workplace policy. All measures will be documented as per local policies. The HAVEN project has had two lay members on the management committee throughout. They have been involved in regular discussions regarding the aims and remit of the HAVEN project.

## DISCUSSION
## Main findings

This study will use structured medical notes review on ward patients misclassified by HAVEN to identify variables that may enhance performance. Any identified variables will be systematically introduced into our score development pipeline to evaluate whether they improve score performance.

## Strengths and limitations of the study

This study is part of a project-wide process to document the development of the HAVEN system such that it is thorough, transparent, repeatable, reportable and the methodology could be useful for other groups developing similar technology.

Unplanned ICU admission is an outcome measure subject to bias, such as the decision-making of individual physicians, local practice guidelines and bed availability.[26 27] This study is limited to one hospital and the results may not be generalisable to other hospitals. Variables identified from the thematic analysis may not be available in the EPR and therefore unable to improve the performance of the HAVEN risk score. Likewise, patients with no previous admissions to the John Radcliffe Hospital will have no available comorbidity data, potentially limiting the performance of the risk score in these patients. To assess the impact of these missing data, we will undertake subgroups analyses in those patients with/ without prior admissions.

While a significant proportion of ICU admissions are referred directly from the emergency department (ED), the HAVEN system was designed specifically for ward patients needing the attention of the critical care team. By excluding these ED referrals, we are reducing the number of eligible patients for this study.

## Implications

To the best of our knowledge, this is the first protocol to describe a study of this type. We hope this protocol will assist future development of similar systems.

**Acknowledgements** JM would like to acknowledge the University of Adelaide, Department of Acute Medicine, who are administering his PhD.

**Contributors** JM and OCR designed the study, undertook the methodological planning and wrote the protocol. DY and PJW assisted in study design and GL commented on successive drafts of the manuscript. All the authors read and approved the final manuscript.

**Funding** This publication presents independent research supported by the Health Innovation Challenge Fund (HICF-R9-524 and WT-103703/Z/14/Z), a parallel funding partnership between the Department of Health and Wellcome Trust.

**Disclaimer** The views expressed in this publication are those of the author(s) and not necessarily those of the Department of Health or Wellcome Trust.

**Competing interests** None declared.

**Patient consent for publication** Not required.

**Ethics approval** Health Research Authority approval was obtained from the South Central Oxford C Research Ethics Committee (16/SC/0264) and the Confidentiality Advisory Group (16/CAG/0066). Informed consent will not be obtained from the patients; however, patients who have requested that their data are not used for research purposes will be identified and removed from the study database. Patients will be allocated a study ID and all data transferred to the research database will have directly identifiable information removed. All documents will be stored securely and only accessible by study staff and authorised personnel. The study will comply with the UK Data Protection Act 2018.

**Provenance and peer review** Not commissioned; externally peer reviewed.

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
