## [Reviewer comments · BMJ Open]

ARTICLE DETAILS

TITLE (PROVISIONAL)	Testing a digital system that ranks the risk of unplanned Intensive Care Unit admission in all ward patients: protocol for a prospective observational cohort study
AUTHORS	Malycha, James; Redfern, Oliver; Ludbrook, Guy; Young, Duncan; Watkinson, Peter

VERSION 1 – REVIEW

REVIEWER	Joseph Tonna University of Utah, Utah, United States
REVIEW RETURNED	21-Jul-2019

GENERAL COMMENTS	Review of: Testing a digital system that ranks the risk of unplanned Intensive Care Unit admission in all ward patients: protocol for a prospective observational cohort study BMJ Open Overall this manuscript reports the study protocol for an observational prospective study to identify predictors of unplanned ICU admission within a hospital system in the United Kingdom. The methodology is clear, as is the language it is written in, though overall the organizational structure isn't as readily familiar as a typical research manuscript. There are a few minor and major comments below for the manuscript and of the overall study design it proposes. Critical: None Major: Page 5, Line 31: You state that the coding typically occurs after discharge, and therefore the patients' comorbidity indices were calculated from previous admissions. This seems to imply that patients with a first hospital admission would have missing data. Is this the case? I can see this being a problem, depending on how it affects the score performance. Page 6, Line 12: Why were Accident/Emergency Department patients not included in the dataset? This cohort will likely contribute significantly to the number of ICU admissions. I see this being a limitation of the study. What about patients admitted to an observational/short stay unit? Minor:
---

	Page 4: Line 50. Please change the positioning of the (within 24 hours) or explain. Does the prediction occur within 24h after admission to the hospital, or rather does it predict admission to the ICU within 24h of score calculation? If the later, which makes the most sense, it should read: “We have developed a system to predict the need for planned ICU Admission within 24 hours among ward patients.” Page 5, line 5: EPR is not defined at first use Page 5, Line 6: Please define and explain “trusts.” This not a term used worldwide. Page 5, Line 30: Please define UK at first use.
--	---

REVIEWER	Luke Keele University of Pennsylvania, USA
REVIEW RETURNED	25-Jul-2019

GENERAL COMMENTS	In this protocol, the investigators are proposing a study to improve an early warning system for ICU admission. I have no objections to the overall study design. In general, I think this is a strong proposal. That is, the investigators will study cases that are misclassified by the statistical model and seek to identify new measures the might improve predictive performance. My only criticism is that here could be a bit more information on how they will proceed if they find new variables. Will these simply be added to the existing model? Or will there be a more through model building step.
---

VERSION 1 – AUTHOR RESPONSE

1. (Reviewer 1) Page 5, Line 31: You state that the coding typically occurs after discharge, and therefore the patients’ comorbidity indices were calculated form previous admissions. This seems to imply that patients with a first hospital admission would have missing data. Is this the case? I can see this being a problem, depending on how it affects the score performance.

We thank the reviewer for highlighting this point. As mentioned, the absence of diagnostic coding prior to discharge could impact the performance of the risk score in patients with known comorbidities (e.g. COPD, chronic heart failure) but with no previous admissions to the study hospital. We now acknowledge this in the manuscript (page 4, lines 23- 24) and we will undertake sub-group analyses by evaluating the score performance in those with and without data from prior admissions (page 6, lines 34 – 36).

2. (Reviewer 1) Page 6, Line 12: Why were Accident/Emergency Department patients not included in the dataset? This cohort will likely contribute significantly to the number of ICU admissions. I see this being a limitation of the study. What about patients admitted to an observational/short stay unit?

The reviewer is correct to highlight that a significant proportion of ICU admissions are referred directly from the Emergency Department (ED). In the UK, there are already robust processes in place by which ED patients can be directly referred to the critical care. In contrast, multiple reports into failures of care have highlighted that patient deterioration on general hospital wards can often go unnoticed. The HAVEN system was designed specifically to bring this cohort of patients to the attention of the critical care team and consequently restrict our evaluation to patients on general wards (page 6, lines 38 – 41).

In our study, observational/short stay units will be treated as any other general ward. We have clarified these points in the text (page 5, lines 7 – 8)

3. (Reviewer 1) Page 4: Line 50. Please change the positioning of the (within 24 hours) or explain. Does the prediction occur within 24h after admission to the hospital, or rather does it predict admission to the ICU within 24h of score calculation? If the later, which makes the most sense, it should read: “We have developed a system to predict the need for planned ICU Admission within 24 hours among ward patients.”

We have amended the manuscript as directed by the reviewer (page 3, lines 41 – 42)

4.(Reviewer 1)

Page 5, line 5: EPR is not defined at first use

Page 5, Line 6: Please define and explain “trusts.” This not a term used worldwide.

Page 5, Line 30: Please define UK at first use.

These have been corrected.

5. (Reviewer 2) My only criticism is that here could be a bit more information on how they will proceed if they find new variables. Will these simply be added to the existing model? Or will there be a more thorough model building step.

We apologise for the lack of clarity here. The output of this study is limited to deriving a list of variables that if used in the risk score, have the potential to improve performance. As the reviewer suggests, we will then introduce these variables into our algorithm development pipeline (using our retrospective data set) to determine whether they improve the discrimination of the risk score. We have briefly clarified how we will pursue this future work within the discussion section. (Page 6, lines 23-24)

VERSION 2 – REVIEW

REVIEWER	Joseph Tonna University of Utah, Utah, United States
REVIEW RETURNED	13-Aug-2019
GENERAL COMMENTS	Comments addressed. Thank you.